# Planctomycetes of the Genus *Singulisphaera* Possess Chitinolytic Capabilities

**DOI:** 10.3390/microorganisms12071266

**Published:** 2024-06-22

**Authors:** Anastasia A. Ivanova, Daniil G. Naumoff, Irina S. Kulichevskaya, Andrey L. Rakitin, Andrey V. Mardanov, Nikolai V. Ravin, Svetlana N. Dedysh

**Affiliations:** 1Winogradsky Institute of Microbiology, Research Center of Biotechnology, Russian Academy of Sciences, Moscow 119071, Russia; ivanovastasja@gmail.com (A.A.I.); daniil_naumoff@yahoo.com (D.G.N.); kulich2@mail.ru (I.S.K.); 2Institute of Bioengineering, Research Center of Biotechnology, Russian Academy of Sciences, Moscow 119071, Russia; rakitin@biengi.ac.ru (A.L.R.); mardanov@biengi.ac.ru (A.V.M.); nravin@mail.ru (N.V.R.)

**Keywords:** *Planctomycetota*, *Singulisphaera*, chitin degradation, glycoside hydrolases, GH18 family, search for homologues

## Abstract

Planctomycetes of the genus *Singulisphaera* are common inhabitants of soils and peatlands. Although described members of this genus are characterized as possessing hydrolytic capabilities, the ability to degrade chitin has not yet been reported for these bacteria. In this study, a novel *Singulisphaera* representative, strain Ch08, was isolated from a chitinolytic enrichment culture obtained from a boreal fen in Northern European Russia. The 16S rRNA gene sequence of this isolate displayed 98.2% similarity to that of *Singulisphaera acidiphila* MOB10^T^. Substrate utilization tests confirmed that strain Ch08 is capable of growth on amorphous chitin. The complete genome of strain Ch08 determined in this study was 10.85 Mb in size and encoded two predicted chitinases, which were only distantly related to each other and affiliated with the glycoside hydrolase family GH18. One of these chitinases had a close homologue in the genome of *S. acidiphila* MOB10^T^. The experimental verification of *S. acidiphila* MOB10^T^ growth on amorphous chitin was also positive. Transcriptome analysis performed with glucose- and chitin-growth cells of strain Ch08 showed upregulation of the predicted chitinase shared by strain Ch08 and *S. acidiphila* MOB10^T^. The gene encoding this protein was expressed in *Escherichia coli*, and the endochitinase activity of the recombinant enzyme was confirmed. The ability to utilize chitin, a major constituent of fungal cell walls and arthropod exoskeletons, appears to be one of the previously unrecognized ecological functions of *Singulisphaera*-like planctomycetes.

## 1. Introduction

Members of the genus *Singulisphaera* are stalk-free planctomycetes with spherical cells, which multiply by budding and may be assembled in short chains or shapeless aggregates [1]. This genus belongs to the family *Isosphaeraceae*, order *Isosphaerales*, class *Planctomycetia* of the bacterial phylum *Planctomycetota*, whose representatives possess complex cell organization, large genomes, and colonize a wide spectrum of natural environments [2,3,4]. Similar to most other described members of the class *Planctomycetia*, *Singulisphaera* species are aerobic, mesophilic, chemoorganotrophic bacteria, which utilize sugars, N-acetylglucosamine, and various biopolymers of plant and microbial origin as growth substrates. Cultured representatives of these planctomycetes from acidic *Sphagnum*-dominated peatlands display mildly acidophilic phenotypes [1,5], while an isolate from a temperate forest soil is a neutrophilic bacterium [6]. The genomes of *Singulisphaera* representatives are the largest among other characterized planctomycetes. Thus, the genome size of *Singulisphaera* sp. strain GP187 is 10.69 Mb [6]. This is the largest genome of all cultured *Isosphaeraceae* strains and the second-largest genome of cultured *Planctomycetota* members [7]. The genome of the type strain of the species *Singulisphaera acidiphila*, MOB10^T^ (=DSM 18658^T^), is 9.76 Mb in size. Genomes of these planctomycetes encode a wide repertoire of carbohydrate-active enzymes (CAZymes), many of which cannot be assigned to any of the currently recognized CAZy families [8]. Thus, *Singulisphaera* species possess high glycolytic potential, which remains to be explored.

As evidenced by the results of cultivation-independent studies, members of the genus *Singulisphaera* are among the most abundant populations of planctomycetes in boreal and subarctic wetlands and soils [9,10,11,12,13]. The use of co-occurrence network analysis identified *Singulisphaera* as one of the key genera that displayed densely connected interactions with many prokaryotic inhabitants of the rhizosphere soil [13]. One of the reasons for this environmental success of *Singulisphaera* species is their ability to utilize a range of plant-derived polysaccharides, such as xylan, lichenan, pectin, laminarin, and some others. What is even more important, *Singulisphaera acidiphila* was identified among primary degraders of exopolysaccharides produced by some soil bacteria by means of stable-isotope probing [14]. The ability to degrade chitin, a major component of fungal cell walls and arthropod skeletons, has not been reported in the original taxonomic descriptions of *Singulisphaera* species [1,5]. Nonetheless, planctomycetes of this genus were found as the part of chitin-degrading microbial community that developed under oxic conditions in agricultural soil slurries supplemented with chitin [15]. Additional evidence for the presence of chitinolytic capabilities in *Singulisphaera* was obtained in the study of mycosphere bacteria associated with *Russula griseocarnosa*, a wild, ectomycorrhizal soil fungus [12]. As shown in this study, the relative abundance of *Singulisphaera* was significantly higher in the mycosphere of this fungus than that in a bulk soil. Thus, the ability to use chitin as a growth substrate and the occurrence of chitinases in *Singulisphaera* species may have been overlooked in previous cultivation-based studies.

Chitinases, or 1,4-β-poly-N-acetylglucosaminidases (EC 3.2.1.14), are endo-glycoside hydrolases catalyzing the cleavage of chitin and chitodextrins [16,17,18,19,20,21,22]. The enzymes with such activity are widespread in nature and are assigned in the CAZy database to four families: GH18, GH19, GH23, and GH48 [23]. A vast majority of experimentally characterized prokaryotic chitinases belong to the family GH18 (clan GH-K). An investigation of phylogenetic relationships among the proteins of this family revealed their complex evolutionary history and demonstrated the tremendous role of a lateral gene transfer [24].

Recently, the capability for chitin degradation was demonstrated for *Fimbriiglobus ruber* SP5^T^, a peat-inhabiting member of the family *Gemmataceae* [7]. The gene encoding the putative chitinase of *F. ruber* SP5^T^ was cloned and expressed in *E. coli*, with subsequent confirmation of chitinolytic activity in the recombinant enzyme. Experimental evidence also confirmed the growth of this planctomycete on amorphous chitin, particularly when this biopolymer served as the sole source of nitrogen rather than carbon [7]. A phylogenetic analysis of *F. ruber* chitinase revealed a close match in the genome of *Planctomicrobium piriforme* P3^T^, a member of the family *Planctomycetaceae*, isolated from a littoral wetland of a boreal lake [20]. Experimental tests confirmed its ability to grow on amorphous chitin. Notably, *P. piriforme* P3^T^ was able to utilize it as both a carbon and nitrogen source. Enzyme activities for chitin degradation, determined through tests with fluorochrome-labeled chito-oligosaccharides, were exclusively associated with planctomycete cells [20].

In this study, a novel *Singulisphaera*-like planctomycete, strain Ch08, was isolated from a chitinolytic enrichment culture obtained from a boreal fen in Northern European Russia. This finding was somewhat unexpected since chitinolytic capabilities have not been described in *Singulisphaera* planctomycetes. The isolation of strain Ch08 opened a possibility to experimentally test the presence of chitinolytic capabilities in *Singulisphaera* species and to perform a detailed phylogenetic analysis of the putative chitinases encoded in the genomes of these bacteria. This was followed by the experimental verification of chitinase activity by means of transcriptomic analysis as well as by cloning and expressing this enzyme in *E. coli*. Apart from *Fimbriiglobus ruber* SP5^T^ and *Planctomicrobium piriforme* P3^T^, this is the third report of chitinolytic potential in members of the phylum *Planctomycetota*.

## 2. Materials and Methods

### 2.1. Source of Isolation and Culture Conditions

Strain Ch08 was isolated from peat soil (pH 7.1) collected in August 2019 from a depth of 10 to 12 cm of the Shichengskoe fen profile (Vologda region, Northern European Russia; 59°56′31.6″ N, 41°15′53.5″ E). The plant community at the sample collection site was represented by an association of *Carex lasiocarpa*, *Carex dioica*, *Eriophorum latifolium*, and *Sphagnum warnstorfii*. The electrical conductivity of fen water ranged from 394 to 408 μS/cm. The total organic carbon content in the peat sample was 73.6%, and the total nitrogen content was 2.3%. A more detailed characterization of the sample collection site was provided in our previous work [25].

The collected peat sample was used to obtain an enrichment culture of microorganisms that participate in hydrolysis of chitin, one of the key biopolymers of wetland ecosystems. For this purpose, 2 g of peat fragmented with sterile shears and 0.1% amorphous chitin, prepared as described elsewhere [26], were placed into 160-mL glass flasks and filled with 30 mL of native fen water. The enrichment culture was incubated at room temperature (20–22 °C) for 4 weeks. The microscopic examination of cell suspensions from this enrichment culture revealed the presence of spherical cells of *Isosphaeraceae* planctomycetes, whose ability to degrade chitin was not known before. The isolation attempt, therefore, was focused on this group of bacteria and involved spread plating of 20 μL aliquots of the enrichment cultures onto medium M31 (modification of medium 31 used for isolation of planctomycetes and described in [27]), solidified with 10 g Phytagel (Sigma-Aldrich, Steinheim, Germany), containing the following (per liter distilled water): 0.1 g KH_2_PO_4_, 20 mL Hutner’s basal salts, 1.0 g N-acetylglucosamine, 0.2 g ampicillin (sodium salt), 0.1 g peptone, 0.1 g yeast extract, pH 6.8. The plates were then incubated at 22 °C for 4 weeks. A number of very small (≤1 mm in size), light-pink-colored colonies that developed on these plates were picked, examined microscopically and taken for further purification procedure. The obtained pure culture, strain Ch08, was maintained on medium M31 and was sub-cultured at 2-month intervals.

### 2.2. Tests for Growth on Chitin and Whole-Cell Hybridization

The ability of strain Ch08 and *Singulisphaera acidiphila* MOB10^T^ (=DSM 18658^T^) to grow on chitin was tested in 160-mL serum bottles with liquid medium containing 20 mL/L of Hutner’s trace element solution, 1 mL/L of vitamin solution, and 0.05 g/L of yeast extract. The sole carbon source was amorphous chitin (0.1%) and the nitrogen source was (NH_4_)_2_SO_4_ (0.01%). In the controls, chitin was excluded. The experiment was carried out in three replicates at the incubation temperature of 22 °C. To determine the cell numbers, suspension aliquots were collected at the beginning of the experiment and after incubation for 10 and 20 days. Planctomycete cells were counted using phase contrast microscopy with a Zeiss Axioplan 2 microscope (Zeiss, Jena, Germany) on Teflon-laminated slides coated with 0.1% gelatin solution. Bacterial abundance in the samples was determined by counting cells in 100 fields of vision for each experimental variant and calculating cell numbers per 1 mL of culture.

In order to enhance the identification of cells associated with chitin particles, Fluorescence in Situ Hybridization (FISH) was additionally conducted as described in our previous studies [7,20]. A combination of two 16S rRNA-targeted planctomycete-specific Cy3-labeled oligonucleotide probes, PLA46 (5′-GACTTGCATGCCTAATCC-3′) and PLA886 (5′-GCCTTGCGACCATACTCCC-3′) [28], was applied for the specific detection of cells on micro-particles of chitin. The oligonucleotide probes were purchased from Syntol (Moscow, Russia). Hybridization was performed on gelatin-coated (0.1%, wt/vol) and dried Teflon-laminated slides (MAGV, Rabenau, Germany) with eight wells for independent positioning of the samples. The fixed samples were applied to these wells, hybridized to the corresponding fluorescent probes, and stained with the universal DNA stain 4′,6-diamidino-2-phenylindole (DAPI; 1 µM). The cell counts were carried out with a Zeiss Axioplan 2 microscope (Zeiss, Jena, Germany) equipped with Zeiss filters 20 and 02 for Cy3-labeled probes and DAPI staining, respectively.

### 2.3. Genome Sequencing and Annotation

DNA for genome sequencing was isolated using the standard CTAB–phenol/chloroform procedure. A part of the obtained genomic DNA was sequenced in an R9.4 cell of a MinION device (Oxford Nanopore, Oxford, UK) using the Ligation Sequencing kit 1D according to the manufacturer’s instructions. The rest of the genomic DNA was sequenced on the Illumina MiSeq platform. Library preparation and sequencing were performed commercially by the ReaGen company (Moscow, Russia). Hybrid assembly of Illumina and Nanopore reads was carried out using the Unicycler software v0.4.9b [29]. The genome sequence of strain Ch08 was deposited in GenBank under accession numbers CP155447 and CP155448 for chromosome and plasmid, respectively. Genome annotation was performed with PROKKA [30]. The genome tree was constructed via multiple alignment of 120 marker genes using the GTDB-Tk program [31]. The analysis also included the genomes of other characterized members of the family *Isosphaeraceae*.

### 2.4. Search for Chitinase-Encoding Genes and Subsequent Phylogenetic Analysis

In this work, we employed the CAZy classification of glycoside hydrolases, which is based on the homology of the amino acid sequences of their catalytic domains [23]. A search for potential glycoside hydrolases among the proteins encoded in the genome of Ch08 was carried out on 27 June 2023 on the dbCAN3 server (https://bcb.unl.edu/dbCAN2/index.php) by means of three algorithms: HMMER:dbCAN, DIAMOND:CAZy, and HMMER:dbCAN-sub [32]. For all proteins identified by at least one of the algorithms and recognized as glycoside hydrolases of the families GH18, GH19, GH23, or GH48, their affiliation with the families was verified manually.

A list of experimentally characterized enzymes for the GH18 family of glycoside hydrolases was compiled based on the information available in the CAZy database (http://www.cazy.org/GH18_characterized.html, accessed on 15 May 2024).

The amino acid sequences of ChiA and ChiB (see Appendix A; XBH03150.1 and XBH02807.1) were used to search for their close homologues in the GenPept database (section “non-redundant protein sequences”) with blastp software (http://www.ncbi.nlm.nih.gov/, accessed on 30 August–15 September 2023). 

Multiple sequence alignment was performed manually using the BioEdit software (https://bioedit.software.informer.com/7.2/, accessed on 20 September 2023) while taking into account the results of pairwise alignments performed with blastp. The results of the multiple alignments (after removal of the most variable sequence fragments) were used to construct phylogenetic trees using the PROTPARS program (Protein Sequence Parsimony method, MP) from the PHYLIP package (https://phylipweb.github.io/phylip/, accessed on 20 October 2023). Moreover, programs SEQBOOT, PROTPARS, and CONSENSE were successively used to derive confidence limits, estimated by 1000 bootstrap replicates for each node.

### 2.5. RNA Isolation and Analysis of Transcriptomic Data

The total RNA was isolated from the bacterial cultures (1 × 10^9^ cells for each sample) grown on medium with chitin or glucose as a source of carbon using the RNeasy Mini Kit (Qiagen, Hilden, Germany), according to the manufacturer’s recommendations. RNA samples were treated with DNase I (7 Kunitz units for each sample) (Qiagen, Hilden, Germany). The bacterial ribosomal RNA was removed using QIAseq Fast Select -5s/16S/23S Kit (Qiagen, Maryland, Germany) according to the manufacturer’s recommendations. The cDNA libraries were prepared using the TruSeq RNA Sample Preparation v2 kit (Illumina Inc., San Diego, USA) according to the manufacturer’s recommendations. Sequencing was performed on the Illumina NovaSeq 6000 platform, generating reads of 100 bp.

Raw transcripts were quality checked and filtered with FastQC and cutadapt [33]. Mapping of transcripts to the genome sequence was carried out with bowtie2 [34] and samtools [35] with subsequent visualization in IGV [36]. The strandness of the reads for the following procedures was determined with RSeQC [37]. Summarization of mapped reads was performed with featureCounts [38]. Differential gene expression analyses were performed in R/Bioconductor (release 3.19) in DESeq2 [39] package applying varianceStabilizingTransformation (vst) function for experiments with one replicate.

### 2.6. Cloning, Expression and Purification of Recombinant Chitinase

PCR primers Ch800_1F (5′-TGGATCCGCGCCCGATCAGAA-3′) and Ch800_1R (5′-TCAAAGCTTCTATTTGCCGCTCG-3′) were designed to amplify the sequence of *chiA* gene. The forward primer was designed to target the mature part of the chitinase ChiA without the N-terminal signal peptide. The *chiA* gene was amplified using PCR; the PCR product was then digested with BamHI and HindIII and inserted into pQE30 (Qiagen, Germany) at the corresponding sites, yielding the plasmid pQE30_Ch800_1.

Plasmid pQE30_Ch800_1 was transformed into *E. coli* DLT1270. Recombinant strain was grown in LB medium supplemented with ampicillin and induced to express recombinant chitinase by adding isopropyl β-D-1-thiogalactopyranoside (IPTG) to a final concentration of 1.0 mM at OD600 of approximately 0.5. The strain was further grown at 30 °C for 16 h. Then, 15 mL of the recombinant cells was harvested using centrifugation at 3.500× *g* for 15 min at 4 °C, washed with 50 mM sodium-phosphate buffer (pH 7.0), and resuspended in 0.7 mL of 50 mM sodium-phosphate buffer (pH 7.5), 0.3 M NaCl, and 5 mM imidazole. The cell extract after sonication was centrifuged (15.000× *g*, 4 °C, 20 min), and the recombinant protein was purified using metal affinity chromatography using a Ni-NTA Spin Kit (Qiagen, Hilden, Germany). Upon elution from the column, the protein (0.4 mL) was dialyzed twice on a Slide-A-Lyzer MINI Dialysis Unit MWCO 3500 (Thermo Scientific, Rockford, USA) against 100 mL of 25 mM sodium-phosphate buffer (pH 7.0) at 4 °C for 1 h. The concentration of the purified protein was determined using the Bradford method using BSA as a standard.

### 2.7. Assay of Chitinase Activity

Chitinolytic activities of the purified ChiA protein were measured in a fluorometric assay with 4-methylumbelliferyl (4-MU) derivatives using a Chitinase Assay Kit (Sigma, CS1030, Saint Louis, USA), according to the manufacturer’s recommendations. The following substrates were used: 4-MU-N-acetyl-β-D-glucosaminide, 4-MU-N, N′-diacetyl-β-D-chitobioside, and 4-MU-β-D-N, N′, N″-triacetylchitotriose. The assays with appropriate substrates (0.2 mg/mL) and ChiA chitinase (4.7 μg/mL) were performed in 0.1 mL for 40 min. The fluorescence of liberated 4-MU was measured using the fluorimeter Fluorat-02-Panorama (Lumex, Saint Petersburg, Russia), with excitation at 360 nm and emission at 450 nm. One unit of activity was defined as the amount of enzyme required to release 1 μmole of 4-MU from the appropriate substrate per minute at 20 °C in 50 mM sodium-phosphate buffer (pH 6.0).

Citrate-phosphate buffer (50mM, pH ranges 3.0–7.0) and sodium-phosphate buffer (50 mM, pH range 6.0–8.0) were used for determining the optimum pH for chitinase activity. The reaction mixtures (0.1 mL) contained 4-MU-N, N′-diacetyl-β-D-chitobioside (0.2 mg/mL) and ChiA chitinase (1.9 μg/mL). The assay was carried out for 20 min. Similarly, chitinase assay was performed at various temperatures (0–50 °C) to determine the temperature optimum of the enzyme.

## 3. Results

### 3.1. Cell Morphology and Identification of Strain Ch08

Strain Ch08 was represented by non-motile spherical cells, 2–3 µm in diameter, which occurred singly, in pairs or in short unstable chains (Figure 1A). These bacteria formed light-pink cell masses when grown on agar medium M31 (Figure 1B). Growth in the liquid cultures was homogenous; no cell aggregates were formed.

The 16S rRNA gene sequence of strain Ch08 affiliated with the planctomycete genus *Singulisphaera* and displayed the closest similarity of 98.2% to the corresponding gene sequence of *Singulisphaera acidiphila* MOB10^T^, which was isolated from an acidic *Sphagnum*-dominated wetland in Northern Russia [1].

### 3.2. Growth on Chitin

Since strain Ch08 was isolated from a chitinolytic enrichment culture obtained from a boreal fen, we tested the presence of chitinolytic capabilities in this planctomycete. In the microscopic analysis of the culture, strain Ch08 incubated for 3 weeks in the medium with amorphous chitin, suggesting its ability to grow on this polymeric substrate. Planctomycete cells developed mainly on chitin micro-particles, actively colonizing their surface (Figure 2A,B).

After 20 days of incubation with amorphous chitin, the number of Ch08 cells increased by more than an order of magnitude (Table 1). This increase in cell numbers was comparable to that in the cultures with glucose as a carbon source. By contrast, no significant changes in the cell number of strain Ch08 were observed in control incubations without a carbon source (Table 1).

### 3.3. Genome Analysis of Strain Ch08

Sequencing of the genomic DNA from strain Ch08 on the Nanopore platform yielded 162,083 reads totaling 4.2 Gb. Subsequently, an additional round of sequencing on the Illumina MiSeq platform generated a total of 1,241,288 paired reads, each with an average length of 150 bp. As a result of hybrid genome assembly, two contigs were obtained with lengths of 10,774,123 and 75,083 base pairs. The latter contig represents a single plasmid in the genome. The content of G+C pairs in the genome was 62.12%. The genome annotation using Prokka predicted ~8400 putative protein-coding sequences, eight copies of the rRNA operon, and 73 tRNA genes. The number of genes on the plasmid amounted to 47. The Ch08 genome contains genes that encode enzymes of the key metabolic pathways of chemoorganotrophic bacteria, including glycolysis, the tricarboxylic acid cycle, the pentose–phosphate pathway, and oxidative phosphorylation. This planctomycete possesses the genome-encoded capability for the synthesis of all amino acids.

Based on the results of comparative 16S rRNA gene analysis, the closest phylogenetic relative of strain Ch08 was *S. acidiphila* MOB10^T^. In silico DNA–DNA hybridization of genome sequences from these planctomycetes revealed 29.2 ± 2.4% similarity. The ANI (average nucleotide identity) value determined for genomes of strain Ch08 and *S. acidiphila* MOB10^T^ was 84.8%. The similarity of Ch08 plasmid to the *Singulisphaera acidiphila* plasmid pSINAC1 was 74% with a coverage of 70%. The phylogenomic tree in Figure 3 displays the position of strain Ch08 in relation to other members of the family *Isosphaeraceae*.

### 3.4. Repertoire of Chitinases Encoded in the Ch08 Genome

A dbCAN3-based search for glycoside hydrolases from the GH18, GH19, GH23, and GH48 families revealed three putative members of the family GH18 (see Appendix A). A manual re-examination showed that one of them (CelA; GenPept, XBH08300.1) was incorrectly annotated using DIAMOND algorithm and in fact belongs to the GH5_1 subfamily, so it should have a glucanase (cellulase) but not chitinase activity. The other two proteins (ChiA and ChiB; GenPept, XBH03150.1 and XBH02807.1, respectively) belong to different GH18 subfamilies showing only a very distant relationship (*E*-value = 0.040). Each of these two paralogues have orthologous proteins encoded in the *Paludisphaera mucosa* Pla2^T^ genome. So, ChiB and MDG3005072.1 (56.45% sequence identity) belong to one subfamily, and ChiA and MDG3002890.1 (60.74%) belong to another one.

The screening of the NCBI database with the GH18 domain of ChiB protein as a query allowed us to reveal its 91 closest homologues; half of them (43 proteins) represented the phylum *Planctomycetota*. All these proteins, as well as 12 experimentally characterized chitinases, were taken for multiple sequence alignment construction and subsequent phylogenetic analysis. The chitinases were selected based on the level of sequence similarity. Planctomycetal proteins comprise four clusters on the phylogenetic tree (Figure 4). One of them (89.5% bootstrap support) consists of representatives of the orders *Gemmatales* and *Isosphaerales* only, while another one (51.2%) includes only members of the *Pirellulales* and *Planctomycetales*. The other two clusters together include only five planctomycetal proteins encoded exclusively in metagenome-assembled genomes.

Screening of the NCBI database with ChiA protein as a query allowed us to find its close relationship with predicted chitinases from *Paludisphaera borealis* PX4^T^ (APW59709.1; 60% sequence identity) and *Singulisphaera acidiphila* DSM 18658^T^ (AGA26805.1; 84%), which had been used as a part of an outgroup in the GH18 family phylogenetic analysis performed by us previously [20]. Thus, we decided to use all 17 proteins from the outgroup of that work for the current phylogenetic analysis. Furthermore, we added the 32 closest homologues based on the NCBI database screening (altogether 50 proteins). Chitinases from *Fimbriiglobus ruber* SP5^T^ (OWK46432.1), *Planctomicrobium piriforme* P3^T^ (SFI04223.1), and 12 other planctomycetal proteins from clusters I and II [20], as well as two non-planctomycetal proteins from cluster II (ATB30918.1, *Myxococcota* and SEH38101.1, *Bacillota*), were added to the phylogenetic analysis as an outgroup. Also, nine experimentally characterized chitinases, selected based on the level of sequence similarity, were taken for phylogenetic analysis (only one representative per bacterial genus was used). Planctomycetal proteins comprised three main clusters on the phylogenetic tree (Figure 5). One of them was very stable (96.6% bootstrap support) and contained all analyzed proteins from members of the *Isosphaerales*, including ChiA from strain Ch08.

### 3.5. Findings Made using Transcriptome Analysis

A total of 395,118,860 single-end sequences were obtained from the glucose-grown culture of strain Ch08. Of these, 393,942,542 sequences were retained after quality filtering and 98.13% of these reads could be aligned to the genome sequence of strain Ch08. Chitin-grown culture of strain Ch08 yielded 459,035,306 reads, of which 457,751,565 sequences were retained after quality filtering. The overall alignment rate was the same as for the glucose-grown culture and the average Phred quality score was 36.

The classical differential gene expression analysis in DESeq2 was not feasible as our dataset included two single replicates from glucose- and chitin-grown cultures. Instead, the vst function, recommended by DESeq2 developers, was employed. In total, 8475 transcripts of protein-coding genes were identified, with 1111 protein-coding genes (13%) showing upregulation (the log fold change (LFC) ≥ 1) and 1667 genes (20%) showing downregulation (LFC ≤ −1) in strain Ch08 during the growth on chitin. To identify functional categories and pathways, these sets of up- and downregulated genes were analyzed using the KEGG database. Only 26.3% of the upregulated genes and 21.8% of the downregulated genes could be annotated by KEGG. The most notable groups of upregulated genes belonged to the categories “Carbohydrate Metabolism”, “Energy Metabolism” and “Nucleotide metabolism” (Appendix A). Downregulated genes fell into the categories “Genetic information processing”, “Signaling and cellular processes”, and “Environmental information processing” (Appendix A).

According to the conducted calculations, the LFC values for the chitinase genes *chiB* and *chiA* were −1.4 and 0.3, respectively. The downregulated expression of the former gene and upregulated expression of the latter one signify distinct regulatory responses of strain Ch08 to applied growth conditions and suggest the presence of chitinase activity for the gene *chiA*. As suggested in our previous study, planctomycete cells may have been attached to chitin micro-particles by means of type IV pili [7]. Notably, in case of strain Ch08, the vast majority of genes involved in pili-mediated attachment, including Flp, TadA/PilB, TadB/PilC, TadC, TadE, TadV/PilD, TadZ, PilT, and RcpC, were upregulated during growth on chitin.

### 3.6. Functional Characterization of the Recombinant Chitinase

The *chiA* gene encodes a 377 a.a. protein comprising the signal peptide and the GH18 catalytic domain. For the functional analysis of the recombinant chitinase, the *chiA* gene fragment coding for the mature protein lacking an N-terminal signal peptide was expressed in *E. coli*. The recombinant chitinase was purified using Ni-NTA affinity chromatography. The chitinolytic activities of the recombinant ChiA protein were evaluated with synthetic soluble substrates.

The chitinolytic activity was maximal with chitobiosidase substrate 4-MU-N, N′-diacetyl-β-D-chitobioside, while an almost ten-fold lower activity was observed with 4-MU-β-D-N, N′, N″-triacetylchitotriose and very low activity was recorded with the β-N-acetylglucosaminidase substrate 4-MU-N-acetyl-β-D-glucosaminide (Table 2).

The chitinase ChiA was active in a narrow pH range with the optimum at pH 6.0. The enzyme had the maximum of chitobiosidase activity at 20 °C and retained more than 50% activity between 5 °C and 35 °C (Appendix A).

### 3.7. Verification of Chitinolytic Capability in S. acidiphila MOB10^T^

The finding of a close homologue of ChiA chitinase from strain Ch08 in the genome of *S. acidiphila* MOB10^T^ (GenPept, AGA26805.1) prompted us to re-examine the occurrence of chitinolytic capability in the latter planctomycete. The results of the growth experiments, which were performed in the same way as with strain Ch08, are shown in Figure 2C,D and Table 1. As seen from this figure and table, *S. acidiphila* MOB10^T^ was clearly capable of growth on chitin as a carbon source. The cell numbers of *S. acidiphila* MOB10^T^ in both chitin- and glucose-grown cultures after 20 days of incubation were approximately twice higher than those of strain Ch08, presumably due to higher specific growth rates displayed by *S. acidiphila* MOB10^T^ on appropriate growth substrates.

## 4. Discussion

This study extends the range of biopolymers degraded by planctomycetes of the genus *Singulisphaera*. According to our previous knowledge, this range included xylan, lichenan, pectin, laminarin, gelatin, pullulan, aesculin, and chondroitin sulfate [1,5]. The newly discovered ability to utilize chitin, a major constituent of fungal cell walls and arthropod exoskeletons, appears to be one of the previously unrecognized important ecological functions of *Singulisphaera*-like planctomycetes. The failure to detect their growth on chitin in former studies, most likely, is explained by the use of commercially available chitin from crab shells which, indeed, does not support the growth of these bacteria. This observation was true for *Fimbriiglobus ruber* SP5^T^ [7] and *Planctomicrobium piriforme* P3^T^ [20] as well.

The ability to degrade and utilize various biopolymers is of key importance for assessing the potential of a newly described microorganism to participate in the cycling of plant-, fungal-, and insect-derived organic matter in nature [40]. Unfortunately, the high importance of tests for the presence of hydrolytic capabilities is underestimated by many microbiologists, who tend to skip time-consuming cultivation experiments. The examination of the currently published taxonomic descriptions of planctomycetes shows that, with very few exceptions, they do not contain information on the ability of these bacteria to degrade biopolymers. Thus, potential ecological functions of many described planctomycetes remain underestimated. As seen from Figure 5, quite a number of described planctomycetes possess genome-encoded chitinases closely related to that in strain Ch08 (ChiA), whose activity was proven in our study by cloning and expressing the corresponding enzyme in *E. coli*. The same is true for the planctomycetal chitinases closely related to that in *Fimbriiglobus ruber* SP5^T^ (see the clade indicated as Cluster I in Figure 5) [20].

The GH18 family of glycoside hydrolases is very large and widespread among living organisms. It includes more than four hundred functionally studied proteins with ten types of enzymatic activities. Unfortunately, the subfamily structure of this family has not been developed in the CAZy classification [23]. Therefore, detailed phylogenetic analysis is required to predict the substrate specificity of the proteins being studied. An important step in this process is the division of the family into more or less monospecific subfamilies. One of these attempts was made by us earlier in the process of annotating potential chitinases encoded in the *Acidisarcina polymorpha* SBC82^T^ genome. This is the only acidobacterium for which the ability to use chitin as a source of nitrogen and/or carbon has been experimentally proven. The genome of this strain encodes six proteins from the GH18 family; each of these proteins can be considered as a representative of an independent subfamily [22]. ChiB protein shows only a very distant relationship with each of them (*E*-value ≥ 0.015). At the same time, ChiA protein is significantly closer to them; particularly, AXC13127.1, AXC15261.1, and AXC10969.1 *A. polymorpha* proteins have an *E*-value ≤ 10^−20^. However, they show less than 30% of sequence identity, suggesting different subfamily membership [41]. That is why all *A. polymorpha* proteins were excluded from the current phylogenetic analysis.

Our previous studies showed the presence of divergent members of the GH18 family in planctomycetes. In particular, the phylogenetic analysis of the GH18 family chitinases from freshwater planctomycetes *Fimbriiglobus ruber* SP5^T^ and *Planctomicrobium piriforme* P3^T^ with the already proven function [7,20] revealed that they are closely related (46.1% sequence identity) and belong to a phylogenetic cluster which also includes 12 other planctomycetal proteins [20]. More distant relationships (different GH18 subfamilies) with five other planctomycetal-predicted chitinases (including proteins from *Paludisphaera borealis* PX4^T^ and *Singulisphaera acidiphila* DSM 18658^T^) were revealed as well [20]. In this study, we have demonstrated that the ChiA protein belongs to the latter subfamily. The nine closest enzymatically characterized proteins (listed at Figure 5) for ChiA are chitinases (EC 3.2.1.14), but they have only 28.20–32.75% sequence identity. This analysis, therefore, allowed only a preliminary assumption about the presence of chitinase activity in ChiA protein. The expression of the corresponding gene clearly confirmed our assumption.

The recently performed annotation of a boreal fen planctomycete *Paludisphaera mucosa* Pla2^T^ genome allowed us to find two paralogous genes encoding putative chitinases from the GH18 family [42]. One of them (GenPept, MDG3002890.1) belongs to the same subfamily as ChiA (Figure 5). The other one (MDG3005072.1) forms a common subfamily with ChiB (Figure 4), wherein both these proteins possess a multidomain structure, having two CBM2 domains. According to the CAZy database [23], CBM2 domains can bind cellulose, chitin, and xylan. Moreover, the twelve closest enzymatically characterized proteins (listed at Figure 4) for ChiB are chitinases (EC 3.2.1.14). These facts allow us to conclude that ChiB is the second chitinase in strain Ch08.

In summary, we were able to experimentally confirm the presence of chitinolytic capabilities in *Singulisphaera* planctomycetes and to characterize the GH18 family chitinase from *Singulisphaera* sp. Ch08. Our study also suggested the occurrence of chitinolytic capabilities in some other earlier described planctomycetes of the genera *Aquisphaera*, *Botrimarina*, *Bythopirellula*, *Paludisphaera*, and *Tautonia* (Figure 5), which may be verified in future studies. The genome size of strain Ch08 (10.85 Mb) exceeds that in *Singulisphaera* sp. GP187 (10.69 Mb, [6]) and, currently, is the largest genome of all cultured *Isosphaeraceae* strains. The exact taxonomic position of strain Ch08 remains to be determined. Based on the low DNA–DNA hybridization value of genome sequences from strain Ch08 and *S. acidiphila* MOB10^T^ (29.2 ± 2.4% similarity) and the ANI value of 84.8%, these two planctomycetes represent two distinct species of the genus *Singulisphaera*. Describing strain Ch08 as a novel *Singulisphaera* species is beyond the scope of this work and will be addressed in the future.

## Figures and Tables

**Figure 1 microorganisms-12-01266-f001:**
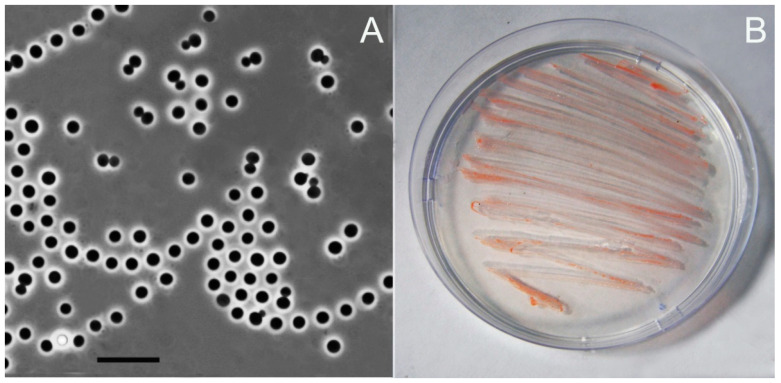
(**A**)—Phase-contrast image of cells of strain Ch08 grown for 14 days on agar medium M31; marker, 5 µm. (**B**)—3-week-old plates with strain Ch08 grown on agar medium M31.

**Figure 2 microorganisms-12-01266-f002:**
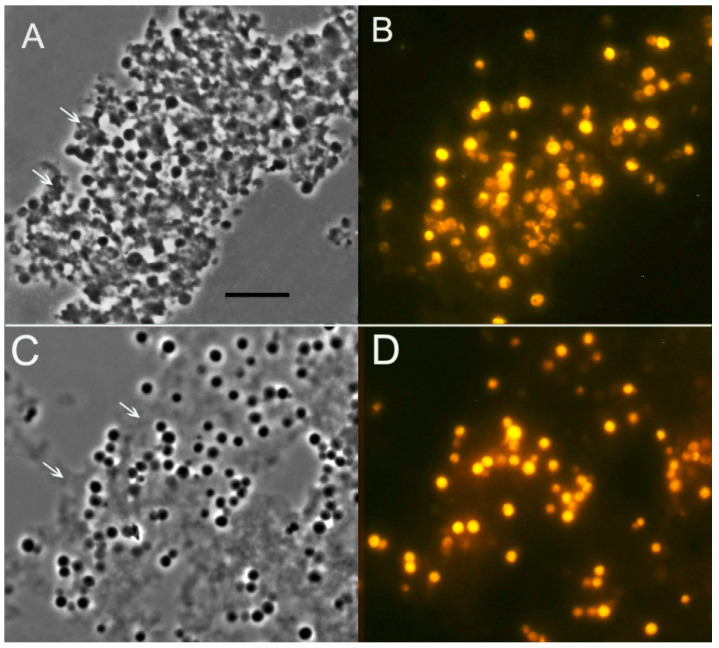
Detection of cells of strain Ch08 (**A**,**B**) and *Singulisphaera acidiphila* MOB10^T^ (**C**,**D**) on micro-particles of amorphous chitin used in growth experiments as a source of carbon. (**A**,**C**)—phase-contrast images, (**B**,**D**)—epifluorescent micrographs of whole-cell hybridizations with Cy3-labeled probes PLA46-PLA886. Bar, 10 µm (applies to all images). White arrows point to chitin micro-particles.

**Figure 3 microorganisms-12-01266-f003:**
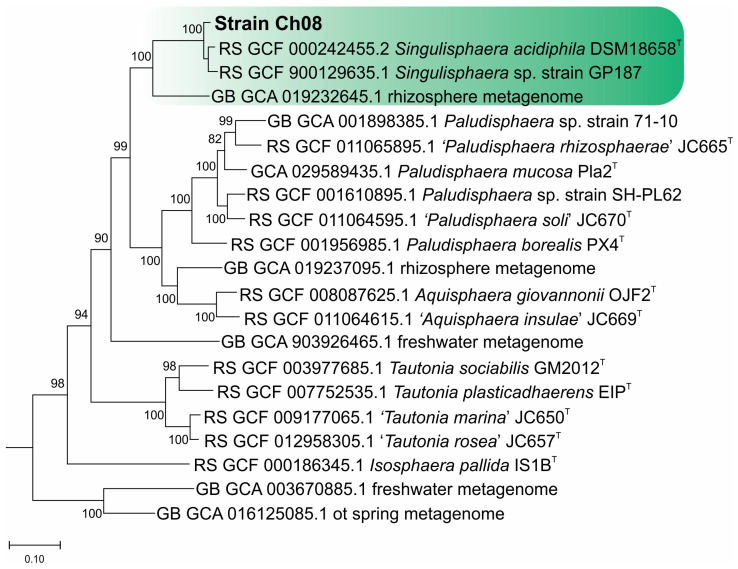
Phylogenomic tree constructed based on comparative analysis of 120 concatenated sequences of conserved marker proteins of strain Ch08, *Singulisphaera acidiphila* MOB10^T^, and other members of the family *Isosphaeraceae*. The genus-level clade of *Singulisphaera* planctomycetes is shown in green. The tree was reconstructed using the Genome Taxonomy Database toolkit, release 2.4.0. The significance levels of interior branch points obtained in maximum-likelihood analysis were determined using bootstrap analysis (100 data re-samplings). Bootstrap values of over 70% are shown. Genomes of anammox planctomycetes available in the GTDB database were used as an outgroup. Bar, 0.1 substitutions per amino acid position.

**Figure 4 microorganisms-12-01266-f004:**
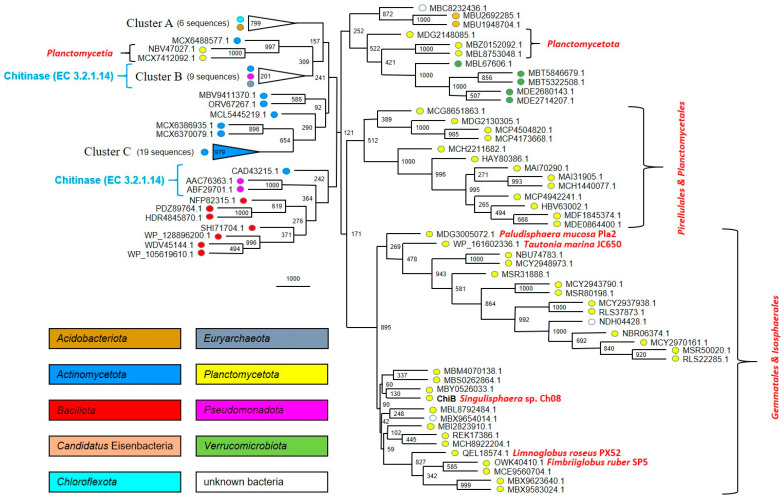
The maximum parsimony phylogenetic tree of the glycoside hydrolase family GH18: ChiB protein (GenPept, XBH02807.1) and its 103 closest homologues. Statistical significance of tree nodes was assessed using bootstrap analysis; the number of supporting pseudoreplicas out of 1000 is indicated at each node. For each of three clusters (A–C), bootstrap support is indicated inside the triangle and the number of proteins is indicated near the triangle. Phylogenetic affiliation of proteins is indicated by colors. Organism names are indicated for cultivated members of the *Planctomycetota* only. The location of 12 enzymatically characterized proteins (all of them are chitinases, EC 3.2.1.14) are indicated: AAL81357.1, AFK20873.1, AFK20874.1, BAA06605.1, BAD85954.1, BAK53891.1, CAA15789.1, CAA62151.1, and CAM00408.1 (Cluster B), as well as AAC76363.1, ABF29701.1, and CAD43215.1.

**Figure 5 microorganisms-12-01266-f005:**
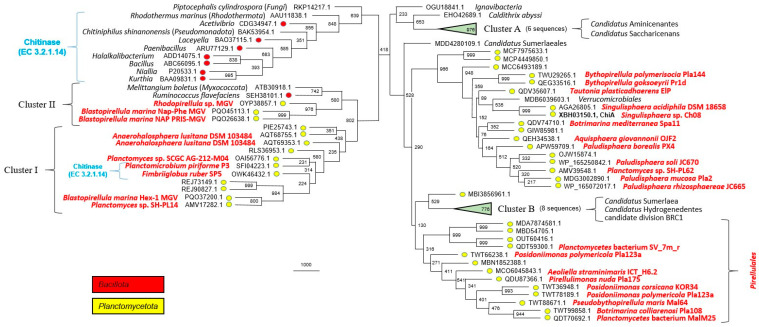
The maximum parsimony phylogenetic tree of the glycoside hydrolase family GH18: ChiA protein (GenPept, XBH03150.1) and its 74 closest homologues. Statistical significance of tree nodes was assessed using bootstrap analysis; the number of supporting pseudoreplicas out of 1000 is indicated at each node. Bootstrap support for clusters A and B is indicated inside the triangle and the number of proteins is indicated near the triangle. Phylogenetic affiliation of proteins to phyla *Bacillota* and *Planctomycetota* is indicated by colors. Organism names are indicated only for cultivated members of the *Planctomycetota*, as well as for all members of other phyla. The location of 11 enzymatically characterized proteins (all of them are chitinases, EC 3.2.1.14) are indicated. Protein clusters I and II from [20] are indicated.

**Table 1 microorganisms-12-01266-t001:** Cell numbers of strains Ch08 and *Singulisphaera acidiphila* MOB10^T^ after 3 weeks of incubation with chitin as a source of carbon. The results of three independent incubations are shown. Values are numbers of cells ml^−1^ (N ± standard errors ×10^7^).

Strains	Cells Number, N × 10^7^ mL^−1^			
Ch08	Days	Experiment (+chitin + NH_4_^+^)	Control (+glucose + NH_4_^+^)	Control (-glucose + NH_4_^+^)
	1	1.5 ± 0.3	1.8 ± 0.5	1.4 ± 0.4
	10	8.3 ± 1.8	12.2 ± 1.9	3.5 ± 0.4
	20	24.3 ± 4.3	38.3 ± 0.5	4.1 ± 0.4
MOB10^T^	1	2.2 ± 0.5	2.4 ± 0.5	1.9 ± 0.9
	10	11.7 ± 1.9	13.2 ± 1.8	3.0 ± 0.7
	20	48.7 ± 9.6	57.0 ± 5.3	2.7 ± 0.5

**Table 2 microorganisms-12-01266-t002:** Specific activity of ChiA chitinase. The results of five measurements are shown (mean ± standard error).

Substrate	Specific Activity (U/mg)	Mode of Activity
4-MU-N-acetyl-β-D-glucosaminide	1.6 ± 0.4	β-N-acetylglucosaminidase
4-MU-N, N′-diacetyl-β-D-chitobioside	2622.1 ± 244.0	chitobiosidase
4-MU-β-D-N, N′, N″-triacetylchitotriose	273.1 ± 33.0	endochitinase

## Data Availability

The raw genomic data are available at GenBank under BioProject number PRJNA1075576. The pools of transcriptome reads obtained for chitin- and glucose-grown cultures of strain Ch08 are available under SRA numbers SRX24890480 and SRX24890479, respectively.

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
