# Peer review of "Planctomycetes of the Genus Singulisphaera Possess Chitinolytic Capabilities"

_microorganisms, 2024, doi:10.3390/microorganisms12071266_

Round 1

Reviewer 1 Report

Comments and Suggestions for Authors

The manuscript by Ivanova et al., described the isolation and characterization of a chitin degrader Singulisphaera sp. strain Ch08. The responsible chitinase-encoding genes were identified using genomics, transcriptomics, and in vitro expression of the chitinase gene followed by protein extraction/purification and enzyme assay. It looks like a comprehensive study of the chitin-degrading strain using a combined physiological and molecular approaches. However, it also looks some important results are missing.

Where is the transcriptomic result? The genes that were significantly up- and down-expressed under the different growth conditions (glucose- VS chitin- growing cells) should be clearly shown as main result. The rest genes expression data should be provided as Supplementary Tables or submitted to a public data base.  

Table 2, replicate and statistical analysis (error bars) are missing.

Did you purify the chitinase from the re-combined E. coli cell? Or did you use the cell-free extract (CFE) for the enzyme assay? If it was the CFE, a negative control should be included. That is, the CFE of E. coli cells with the empty vector should be assayed for chitinase activity.

Author Response

Comment: Where is the transcriptomic result? The genes that were significantly up- and down-expressed under the different growth conditions (glucose- VS chitin- growing cells) should be clearly shown as main result. The rest genes expression data should be provided as Supplementary Tables or submitted to a public data base.

Response:  The reads obtained in the transcriptomic analysis for chitin- and glucose-grown cells of strain Ch08 are now available under SRA numbers SRX24890480 and SRX24890479, respectively. We have extended the corresponding results section 3.5 and also included the Supplementary figure S1 showing up-regulated and down-regulated genes distributed to various KEGG categories in strain Ch08 during growth on chitin.

Comment: Table 2, replicate and statistical analysis (error bars) are missing.

Response: The assays were performed in five replicates. We have added this information in Table 2, as well as standard error values.

Comment: Did you purify the chitinase from the re-combined E. coli cell? Or did you use the cell-free extract (CFE) for the enzyme assay? If it was the CFE, a negative control should be included. That is, the CFE of E. coli cells with the empty vector should be assayed for chitinase activity.

Response: Enzymatic activity was measured for purified recombinant protein, as described in Methods (sections 2.6 and 2.7) and Results (section 3.6). We did not use CFE.

Reviewer 2 Report

Comments and Suggestions for Authors

The manuscript microorganisms-3057152, entitled “Planctomycetes of the genus Singulisphaera possess chitinolytic capabilities”, addresses an important paper showing the discovery of chitinolytic capabilities in Singulisphaera planctomycetes, a genus of bacteria previously not known to degrade chitin. However, in my opinion this paper must be revised in a major manner for reasons of forms and content.

Emphasize the novelty of chitinolytic capabilities in Singulisphaera in the Introduction section.

Briefly mention the potential applications of planctomycete chitinases in the Introduction section.

Consider adding a brief justification for the chosen growth medium components for culturing Ch08 (M31 medium).

Briefly describe the fluorescent chitin-oligosaccharide staining method used for FISH analysis.

Comments on the Quality of English Language

There are some minor errors in grammar and punctuation throughout the manuscript (e.g., missing articles, incorrect verb tenses). It would be helpful to have a proofreader review the manuscript for these errors.

Author Response

Comment: Emphasize the novelty of chitinolytic capabilities in Singulisphaera in the Introduction section.

Response: Done. We have added some text at the end of Introduction section.

Comment: Briefly mention the potential applications of planctomycete chitinases in the Introduction section.

Response: We don’t think that planctomycete chitinases may have any applied potential. Planctomycetes are slow-growing bacteria. Culturing and laboratory maintenance of planctomycetes require patience and high expertise. Many other fast-growing bacteria, which are easy to handle, represent much better sources of chitinases with applied potential. We would like to avoid any unsubstantiated speculations in our manuscript.

Comment: Consider adding a brief justification for the chosen growth medium components for culturing Ch08 (M31 medium).

Response: Done.

Comment: Briefly describe the fluorescent chitin-oligosaccharide staining method used for FISH analysis.

Response: Sorry, this is a clear misunderstanding. FISH analysis applied in our study involves using 16S rRNA-targeted fluorescent probes to detect and enumerate cells of planctomycetes that develop on chitin micro-particles. This method has nothing to do with the fluorescent chitin-oligosaccharide staining method. To make it clear to the reader, we highlight the use of 16S rRNA-targeted fluorescent probes in section 2.2 now.

Comment: There are some minor errors in grammar and punctuation throughout the manuscript (e.g., missing articles, incorrect verb tenses). It would be helpful to have a proofreader review the manuscript for these errors.

Response: We have made one additional round of proofreading. Hope it looks better now.

Reviewer 3 Report

Comments and Suggestions for Authors

<Planctomycetes of the genus Singulisphaera possess chitinolytic  capabilities>,Presents new information on bacteria that  degrade chitin  not reported before:  novel Singulisphaera representative, strain Ch08 from Rusia a  common inhabitants of soils and peatlands.The ability to utilize chitin, (major constituent of fungal cell walls and arthropod exoskeletons), was previously an unrecognized ecological function.

The ability to utilize chitin, a major constituent of fungal walls and arthropod exoskeletons, was previously of  unrecognized ecological function, which can be used for biological control? 

I consider that the impact for use these bacteria and its The ability to utilize chitin, a major constituent of fungal cell  walls and arthropod exoskeletons, appears to be one of the previously unrecognized ecological functions  could be better mentioned. thus potential uses can be more explained.

The paper is well presented and contains illustrative images. The importance of <Planctomycetes of the genus Singulisphaera possess chitinolytic  capabilities>,Presents new information on bacteria that  degrade chitin  not reported before:  novel Singulisphaera representative, strain Ch08 from Rusia a  common inhabitants of soils and peatlands.The ability to utilize chitin, (major constituent of fungal cell walls and arthropod exoskeletons), was previously an unrecognized ecological function.

The ability to utilize chitin, a major constituent of fungal walls and arthropod exoskeletons, was previously of  unrecognized ecological function, which can be used for biological control? 

I consider that the impact for use these bacteria and its The ability to utilize chitin, a major constituent of fungal cell  walls and arthropod exoskeletons, appears to be one of the previously unrecognized ecological functions  could be better mentioned. thus potential uses can be more explained.

The paper is well presented and contains illustrative images. The importance of  of its properties. could be more discused. 

Author Response

Comment: The ability to utilize chitin, a major constituent of fungal walls and arthropod exoskeletons, was previously of  unrecognized ecological function, which can be used for biological control?

Response: We don’t think so. Efficient biological control implies using fast-growing and fast-acting microorganisms, which possess high colonizing capabilities. By contrast, planctomycetes are slow-growing bacteria that are quite difficult in handling. In our opinion, they are not suitable for the use in biotechnologies.

Comment: I consider that the impact for use these bacteria and its The ability to utilize chitin, a major constituent of fungal cell  walls and arthropod exoskeletons, appears to be one of the previously unrecognized ecological functions  could be better mentioned. thus potential uses can be more explained.

Response: See our response to the previous comment.

Round 2

Reviewer 2 Report

Comments and Suggestions for Authors

I am writing to follow up on my review of manuscript microorganisms-3057152, entitled : Planctomycetes of the genus Singulisphaera possess chitinolytic capabilities. I'm pleased to inform you that the authors have addressed all my comments thoughtfully and respectfully.

Their revisions demonstrate a clear understanding of the points I raised, and the changes they have made significantly improve the quality of the manuscript. I believe the paper is now much stronger and ready for further consideration.